# Risks of Solid and Lymphoid Malignancies in Patients with Myeloproliferative Neoplasms: Clinical Implications

**DOI:** 10.3390/cancers12103061

**Published:** 2020-10-20

**Authors:** Mette Brabrand, Henrik Frederiksen

**Affiliations:** 1Department of Haematology, Odense University Hospital, 5000 Odense, Denmark; Mette.Brabrand@rsyd.dk; 2Academy of Geriatric Cancer Research (AgeCare), Odense University Hospital, 5000 Odense, Denmark; 3Department of Clinical Research, University of Southern Denmark, 5000 Odense, Denmark; 4Department of Clinical Epidemiology, Aarhus University Hospital, 8200 Aarhus, Denmark

**Keywords:** myeloproliferative neoplasms, essential thrombocythemia, polycythemia vera, primary myelofibrosis, lymphoma, solid cancer, epidemiology, prognosis, comorbidity, review

## Abstract

**Simple Summary:**

Patients with chronic myeloproliferative neoplasms (MPNs) such as polycythemia vera and essential thrombocythemia have an elevated risk of acute leukemia. Recently, it has been recognized that the risk of solid cancers is also increased. In the past decade, several studies have compared cancer frequency in patients with MPNs with the general population. In our study, we present results sampled from 12 previous studies, totaling more than 65,000 patients with MPNs identified through large registries. Patients with MPNs were compared to the age/sex-matched general population. Our results show that risk of new cancers is 1.5–3.0-fold elevated in patients with MPNs. In particular, lymphomas and cancers of the skin, lung, kidney, and thyroid gland occur more frequently. The difference in cancer occurrence is highest in the age group 60–79 years. Our results indicate that clinical follow up of patients with MPNs should include awareness of the increased cancer risk.

**Abstract:**

In the past decade, several studies have reported that patients with chronic myeloproliferative neoplasms (MPNs) have an increased risk of second solid cancer or lymphoid hematological cancer. In this qualitative review study, we present results from studies that report on these cancer risks in comparison to cancer incidences in the general population or a control group. Our literature search identified 12 such studies published in the period 2009–2018 including analysis of more than 65,000 patients. The results showed that risk of solid cancer is 1.5- to 3.0-fold elevated and the risk of lymphoid hematological cancer is 2.5- to 3.5-fold elevated in patients with MPNs compared to the general population. These elevated risks apply to all MPN subtypes. For solid cancers, particularly risks of skin cancer, lung cancer, thyroid cancer, and kidney cancer are elevated. The largest difference in cancer risk between patients with MPN and the general population is seen in patients below 80 years. Cancer prognosis is negatively affected due to cardiovascular events, thrombosis, and infections by a concurrent MPN diagnosis mainly among patients with localized cancer. Our review emphasizes that clinicians caring for patients with MPNs should be aware of the very well-documented increased risk of second non-myeloid cancers.

## 1. Introduction

Myeloproliferative neoplasms (MPNs) encompassing essential thrombocythemia (ET), polycythemia vera (PV), primary myelofibrosis (PMF), chronic myeloid leukemia (CML), and unspecified MPNs are chronic stem cell cancers of the bone marrow [1]. Although MPNs are considered to be relatively indolent hematological cancers, patients suffer a reduced quality of life, increased risks of vascular complications, MPN progression, leukemic transformation, and consequently reduced life expectancy [2,3,4,5,6,7,8]. Patients with MPNs therefore often remain in life-long clinical follow up to monitor and treat the disease and to avoid or manage complications. 

In the past decade, several large-scale registry studies have also shown that solid cancers and hematological lymphoid malignancies occur more frequently in patients with MPNs compared to the general population [9,10,11,12,13]. The mechanisms behind this increased second cancer risk are elusive but an inherent tendency to cancer, shared risk factors, and anti-neoplastic treatment of MPNs may be involved [4,14,15,16,17,18,19,20,21]. 

Although the elevated risk of non-myeloid second cancers is now well established, the optimal way to include this knowledge in management of patients remains unknown. Physicians caring for patients with MPNs may even consider cancer surveillance programs in asymptomatic patients for specific MPN subtypes and specific cancers. The focus of the current study is to review results from studies reporting on absolute and relative increased risk of non-myeloid second cancer in patients diagnosed with MPNs as their first cancer. Our goal is to facilitate clinical follow up and inform shared decision making by providing absolute and relative second cancer risks and to address whether specific surveillance for solid cancer and lymphoid malignancies is warranted.

## 2. Results

The 12 studies identified included analysis of more than 65,000 patients with MPNs and various population comparison groups. Three studies identified and followed patients through the Swedish cancer registry that were diagnosed with MPN in the period 1958–2006 [22], 1973–2009 [11], or 1958–2015 [13] and therefore some patients are likely to have been reported in multiple publications and analyses. Most of the studies reported on the observed number of patients with second cancer and compared this to the expected number based on the age/sex-standardized incidence ratio (SIR) or incidence rate ratios (IRRs) in the general population [10,11,13,22,23,24,25,26]. Some studies also compared the cumulative incidence of cancer five and ten years after MPN diagnosis, treating death as a competing event [9,11,23]. Most studies included the main subtypes of MPNs, e.g., ET and PV. However, some studies also included patients with primary or secondary myelofibrosis, unspecified MPN, and a single study also included patients with chronic myeloid leukemia (Table 1 and Table 2). All studies were retrospective cohort studies and two studies also included a matched comparison cohort [11,27].

### 2.1. Cumulative Incidence of Solid Cancer in Patients with MPN

Two large studies from the US followed 20,250 and 3941 patients with different MPN subtypes or patients with PV, none of whom were diagnosed with cancer prior to MPN [9,23]. These studies found that the cumulative incidence of solid cancer was 7.9–8% after five years and between 12.7% and 13.1% after 10 years [9,23]. In patients ≤ 60 years with PV, the cumulative incidence of second cancer was 5.0% and 11.0% after five and ten years, respectively; and 9.9% and 14.7% among patients over 60 years [23]. In line with these results, our own study showed that the five-year cumulative incidence of solid cancer was 7.8% in patients with ET, 6.8% in patients with PV and 4.7% in patients with chronic myeloid leukemia (CML) [27]. Compared to the general population, one study computed the absolute excess risk (AER) of cancer as the difference between observed and expected numbers per 10,000 person years and found that the AER was 20.7 for solid cancer [23]. Five-year cumulative cancer incidences were also reported in a study from Sweden including 9379 patients with MPNs and four age- and sex-matched controls per patient from the general population. In this study, the five-year cumulative cancer incidence ranged approximately 2–9% in different age groups of patients with MPNs and was generally 1−2% points lower in the general population [11]. As expected, the cumulative incidence of solid cancer increased with age. However, among patients aged 80+, there were no differences in cumulative cancer incidence attributable to MPN diagnosis [11].

### 2.2. Relative Increase in Cancer Risk

In Table 1, results from studies reporting on risks of solid cancers in patients with MPNs compared to the general population are displayed. As shown, most studies found SIRs or IRRs that correspond to an 1.2- to 1.4-fold general increase in the incidence of solid cancers. Two of the studies also found that even the SIRs are very similar between the different MPN subtypes [9,11], whereas our own study revealed that SIRs were lowest in patients with ET, higher in PV, and highest in CML (Table 1) [10]. During follow up, the studies reported different results on the relative estimate of cancer risk. We found that for patients with ET, SIRs for cancer incidence diminished during the first 5 years following diagnosis, whereas the opposite was seen in patients with PV and CML [10]. Shresta and colleagues reported on insignificant increases in SIR for solid cancer two years after ET diagnosis [28]. Landtblom et al. showed that HRs for any solid cancer and for non-melanoma skin cancer decreased steeply during the first year and gradually increased thereafter [11]. This latter study suggested that some of the observed early increases in cancer frequency probably reflect coincidental findings from routine diagnostic work up during the MPN diagnostic process. 

### 2.3. Specific Solid Cancer Types 

The risk of different solid cancer types in patients with MPN varied remarkably. In all studies, risk of colon cancer was either slightly reduced or identical to that of the general population. This was also the case for breast cancer and prostate cancer—the overall frequency in patients with MPNs was the same as that in the rest of the population (Table 1). For lung cancer, almost all studies found that the frequency was 1.3- to 1.9-fold elevated (Table 1). For melanoma, other skin cancers, as well as thyroid and kidney cancer, the relative cancer incidence was 2−3-fold elevated in almost all studies (Table 1).

### 2.4. Lymphoid Malignancies

For lymphoid malignancies, patients with MPNs have a remarkable and consistent increased risk. This applies both to MPN patients in general, with the relative increase in lymphoid hematological malignancies ranging from 2.8 to 3.4 (Table 2), as well as subtype-specific increases. Patients with ET, PV, or PMF all had relative increases in lymphoid hematological malignancies—both lymphomas and multiple myeloma (Table 2). For specific lymphoid malignancy subtypes such as non-Hodgkin lymphoma, Hodgkin lymphoma, chronic lymphocytic leukemia, and multiple myeloma, the risks were elevated remarkably in patients with MPNs compared to general population comparisons. Although some estimates were imprecise, the general relative increase ranged between a 1.3- and 3.5-fold elevation (Table 2).

### 2.5. Prognosis of Patients with MPN and a Second Cancer

A single paper studied prognosis of cancer patients with MPN compared to age/sex-matched patients with the same cancers but without preceding MPNs [27]. In Table 3, the main results from this study performed by our group published in 2015 are shown. The study included 1246 patients with MPN and a second cancer compared to 5155 comparisons [27]. The study showed that the hazard ratio for death was 1.5-fold elevated for cancer patients with preceding ET, 1.2-fold elevated for cancer patients with PV, and 1.2-fold elevated for cancer patients with CML compared to cancer patients without these myeloproliferative neoplasms prior to their solid cancer (Table 3). Mortality in the first five years was 1.7-fold elevated in cancer patients with ET, 1.5-fold elevated in cancer patients with PV, and 2.3-fold elevated in cancer patients with CML but also remained higher after this time point [27]. The study also showed that the excess risk of death in patients with preceding MPNs was mainly seen in patients with localized cancer or low comorbidity. Survival in cancer patients with either regional spread or distant metastasis was the same as in cancer patients with MPNs who also had regional spread or metastatic second cancer. Of note in cancer patients with CML, the excess mortality was mainly seen in the period prior to the introduction of tyrosine kinase inhibitor treatment from approximately 2000 [27]. After this time point, patients with CML and second cancer had the same survival as cancer patients without preceding CML. Causes of death in cancer patients with MPNs were mainly elevated due to cardiovascular and thromboembolic events with a five-year mortality rate (MR) per 1000 person years of MR_ET_ = 2.2 (95% CI: 1.2–4.2) and MR_PV_ = 1.4 (95% CI 1.0–1.9). However, infection and bleeding also contributed to mortality more frequently in patients with MPNs and cancer than in other cancer patients [27].

### 2.6. Effects of Age on Second Cancer Risk

All studies found that although the absolute number of MPN patients with a second cancer increased with age, the increase relative to the general population was largest in the youngest age groups, reaching a 1.8- to 4.2-fold elevation [9,10,11,23,28]. For example, in the study by Chattopadhyay et al., the relative increase in second cancer risk was highest in MPN patients aged ≤ 65 years, with IRRs ranging from 1.7- to 3.7-fold elevated in patients with ET, PV, MF and MPN-U, whereas the IRRs ranged from 1.3- to 1.8-fold elevated in patients aged > 65 years [13]. The same tendency was seen in other studies, with SIRs for second cancer of 1.8 for PV patients aged ≤ 60 years and 1.2 in patients aged > 60 years [23]; or ET patients with SIRs of 1.8 and 1.2 for ages ≤ 60 years and > 60 years, respectively [28]. In the study by Landtblom et al., the overall cumulative incidence of cancer increased with increasing age for both patients and matched comparisons [11]. The cumulative cancer incidence remained elevated for patients with MPNs across all age groups; however, this difference disappeared in the age group ≥80 years [11]. Similarly, another registry study compared cause-specific death in 9285 MPN patients with 35,000 matched population controls and found that HRs for death due to solid cancer were 1.1–2.5-fold elevated for MPN patients below 80 years but 1.0 in older patients [30]. 

### 2.7. Other Factors Potentially Associated with Second Cancer Risk

In most of the included studies, data were based on routine registrations, and detailed clinical or para-clinical information such as genetic aberrations and treatment were not usually available. However, two studies included different MPN treatments [25,29] or JAK2V617F mutational status [25] in their analyses but did not find them to be associated with second cancer risk. Similarly, the period of MPN diagnosis was also not associated with second cancer risk in three other studies [10,11,13].

## 3. Discussion

Our review study shows that patients with MPNs are at increased risk of solid cancer and lymphoid malignancies. Results from the included studies published in the past decade totaling analysis of more than 65,000 patients with MPNs reveal remarkably consistent results, with approximately 1.5- to 3.0-fold elevated cancer incidence. Noteworthy, the excess solid cancer risk derives mainly from increases in incidences in risks of skin cancer, lung cancer, thyroid cancer and kidney cancer. However, not all cancer types show an increase in frequency. Major solid cancer types such as breast cancer, prostate cancer and colorectal cancer occur as frequently in patients with MPNs as in the general population. In addition, lymphoid malignancies occur approximately 2.5- to 3.5-fold more frequently in patients with MPN as their first cancer compared to the general population. Age seems to modify the cancer risk uniformly across studies, with the largest relative increase in age groups < 60 years. The different subtypes of MPNs have similar relative increases in risk of a second cancer. However, the first assigned MPN subtype diagnosis may change when all clinical and para-clinical data become available or if disease progresses. This means that a first recorded MPN subtype diagnosis may not be accurate and the estimated risks of second cancer attributed to specific MPN subtypes may be affected by this uncertainty. Overall, it remains that regardless of the subtype, patients with MPNs have an increased risk of second cancer—both solid cancers as well as myeloid and lymphoid hematological neoplasms.

The mechanisms behind the elevated risk of second cancer in patients with MPNs are elusive. However, estimated cancer occurrence may derive from increased clinical surveillance as well as shared environmental risk factors (e.g., drugs) and shared genetic risk factors (e.g., mutations). Since patients with MPNs remain in clinical follow up, a surveillance or detection bias for cancer may be considered. Results from one of the Swedish registry studies suggest that detection bias may to some degree influence results [11]. In this study, solid cancer risk was highest in the period immediately after the MPN diagnosis and dropped thereafter. However, the HRs were never uniform at any time point and after approximately five years, HRs showed a continuous rise in line with our own results [10,11]. Noteworthy, our review revealed that major cancer types such as colorectal cancer, breast cancer, prostate cancer as well as other cancer types are not in excess among patients with MPNs and detection bias is therefore unlikely to explain all the observed increased risks. Similarly, in another previous study including patients with ET, an increase in cancer incidence was first seen approximately four years after ET diagnosis, thus also indicating that detection bias does not explain all the observed excess risk for second cancer [21]. 

The drugs used in the management of MPN may influence the risk of transformation into acute myeloid leukemia (AML). However, it is not well known whether the drugs also increase risks of solid cancer or lymphoma. Treatments no longer in general use such as ^32^P and pipobroman induce a well-established increased risk of AML [17,31]. The same applies to busulfan and other alkylating drugs, whereas there is ongoing controversy over whether hydroxyurea may also contribute to this increased risk [15,21,32]. Recent studies also suggest that risks of solid cancer or lymphoid malignancies may be associated with MPN treatment. Within the European Leukemia Network, a case-control study included 647 patients with MPN and a second cancer and 1234 patients with MPN free of other cancers who were matched on age, sex and MPN subtype [20]. This study suggested that exposure to pipobroman, ruxolitinib, or hydroxyurea increased risk of non-melanoma skin cancer, with odds ratios ranging from 2.3 to 3.9, whereas the same was not seen for alpha-interferon, busulfan, or anagrelide exposure [20]. In another study including 196 patients with MPNs, 18.9% were diagnosed with a solid cancer after a median of 6.8 years [33]. In this particular study, approximately half of the diagnosed solid cancers were skin cancers and were seen mainly in patients who were exposed to hydroxyurea [33]. Of note, the hydroxurea-exposed patients were approximately 10 years older than patients who were treated with alpha-interferon, which may have confounded this observation [33]. In another cohort study, 1026 patients with ET without previous malignancies were followed for a median of 6.2 years for second cancer outcomes excluding basal cell skin cancer and acute leukemia [34]. Patients were stratified according to their drug exposure into the following five groups: 1. not treated, 2. hydroxurea treated, 3. alkylating agents (mainly pipobroman), 4. hydroxurea + alkylating agents, and 5. treatment with anagrelide or alpha interferon [34]. Importantly, untreated patients or patients treated with anagrelide or interferon were younger and the drug exposure time was similarly longer in groups treated with alkylating agents. Despite these potential confounding differences, no differences in the cumulative incidence of second cancer across treatment groups were observed [34]. The only patient characteristics that were associated with a second cancer were male sex and age over 60 years. These findings are possibly influenced by inclusion of only the MPN subtype ET, which has been found to have the lowest second cancer risk in some studies, but note that the two main second cancers—skin cancer and acute leukemia—were excluded as outcomes [10,25,26]. Recently, a study including two cohorts of 626 and 929 patients with MPN observed that patients treated with JAK1/2 inhibitors developed aggressive lymphomas (*n* = 6) 15–16-fold more frequently than patients not exposed to these treatments [35]. Due to the small numbers of patients who developed lymphoma, risk estimates were imprecise [35].

In MPN, genetic signatures using mutational status in 69 genes is associated with myeloid disorders predict risk of MPN progression, AML transformation, and survival [4,19]. Similarly, some studies suggest that second cancer risk may increase with MPN subtype severity [10,25,26]. It is therefore conceivable that the generally increased risk of second cancer could in part reflect genetic instability in MPN patients [19]. Interestingly, it has been observed that phenotypically unrelated cancers such as T cell lymphomas of the angioimmunoblastic subtype and MPNs share some genetic markers such as TET2 and DNMT3a mutations that are usually considered myeloid [4,36]. However, in the recent previously mentioned case-control study including cases with MPN and second cancer and age/sex/MPN type-matched controls without a second cancer, no specific mutation signatures were observed in cases vs. controls, indicating that the commonly found mutations in MPNs do not alone advance second cancer risks [20]. Another frequent hypothesis in second cancer pathophysiology among patients with MPN is that chronic inflammation and loss of tumor immune surveillance increase cancer risk [37,38,39].

The elevated risk of lung cancer in patients with MPN deserves special consideration. Smokers and patients with lung disease may develop blood test results that may resemble test results from untreated patients with MPNs such as high hematocrit, thrombocytosis and leucocytosis [40]. If such patients who do not fulfil MPN criteria are erroneously registered with a MPN diagnosis, an elevated lung cancer risk finding will be biased. Yet, smoking has in itself been reported to be associated with increased risk of MPN development, with HRs of 2.5 during a mean of 6.8 years follow up for daily smokers. Additionally, both occurrence of JAK2 and CALR mutations in the general population and the MPN disease have also been found to be associated with smoking in three independent population studies in Denmark [18,41,42]. One of these recent studies even showed that pneumonia and respiratory mortality were elevated in patients with MPNs and that this effect was similar for never smokers and ever smokers, emphasizing that respiratory morbidity, smoking, and a MPN diagnosis are mutually associated [42]. It is unsolved whether the MPN disease per se associates with lung cancer directly or whether it is through shared risk factors with smoking. Noteworthy, our own study demonstrated that cancer risk remains elevated even among PV patients free of chronic pulmonary disease and this may suggest that the MPN disease could convey an independent lung cancer risk [10]. Regardless of the underlying mechanisms, this knowledge may have implications for patient care and follow up and clinical focus on patients’ previous and current smoking habits remains important.

Low-dose aspirin treatment is almost universally used to prevent thrombosis in patients with MPNs [43]. Low-dose aspirin treatment associates with a generally decreased risk of colorectal cancer, possibly via an increase in lower gastrointestinal bleeding and therefore increased detection and removal of colorectal polyps [44,45,46]. It is unknown whether increased detection and removal of polyps also influences risk of colorectal cancer in patients with MPN, where risks are either on par with the general population or decreased. Intriguingly, it has also recently been suggested that low-dose aspirin may also associated with a reduced risk of female genital tract cancer and breast cancer in patients with MPNs [47].

The consequences for second cancer development in patients with MPNs is detrimental. Although one of the Swedish registry studies suggests that solid cancer may be as frequently registered as cause of death among patients with MPN as in the general population, our own study showed that survival is significantly poorer for cancer patients who also have MPN compared to other cancer patients [27,30]. Intriguingly, a recent study suggests that active MPN treatment may negatively affect survival in second cancer. However, the mechanism, e.g., suboptimal second cancer management, is unknown [48].

Causes of death in patients with MPNs and second cancer have mainly been attributed to cardiovascular events, thrombosis and infections, all of which are preventable events [27] Furthermore, the excess mortality was mainly seen in MPN + second cancer patients with localized second cancer—patients who are often eligible for curative treatment [27]. Therefore, supportive cancer care such as thrombosis prophylaxis and infectious prophylaxis is particularly important among cancer patients who already have a MPN diagnosis.

### Routine Cancer Surveillance in Patients with MPNs

Whether patients with MPNs are eligible for routine cancer surveillance remains an open question. Overall, the cumulative incidence of new primary cancers in this group of patients ranges from 2% to 13% in five years and is possibly only 1% to 2% points higher than the general population. Additionally, a single study has shown that the absolute excess solid cancer occurrence in patients with PV is 20.7 cases of solid cancer and 0.65 cases of lymphoma per 10,000 person years of follow up [23]. It therefore seems unlikely that screening programs or routine cancer surveillance among asymptomatic MPN patients will provide a survival benefit [49]. There are, however, subgroups in whom specific procedures or extra surveillance may be considered. The largest absolute difference in cumulative cancer incidence between patients with MPN and MPN-free controls is in the 60 to 79 year old group, indicating that focus on cancer symptoms during follow up is particularly relevant in this age group. Since a large proportion of the excess solid cancers are skin cancers, systematic skin inspection in patients with MPNs at regular intervals should be considered. Further, kidney cancer and lung cancer occur more frequently in patients with MPNs and relevant imaging should be provided at clinical suspicion, particularly among high-risk patients, for these particular cancers. 

## 4. Materials and Methods 

The relevant literature for this qualitative review was identified via the Pubmed database using the search term: myeloproliferative neoplasm AND (new primary cancer OR second cancer) published after 1 January 2000. Our literature search was conducted on 23 April 2020 and resulted in 1426 hits (Figure 1). Additionally, eight potentially relevant articles were identified through other sources such as reference lists and own work. All identified articles were screened by title, leaving 26 articles that qualified for full-text review (Figure 1). Of these, 12 articles reported on cancer occurrence after a MPN diagnosis compared either to matched general population controls or to cancer incidence statistics from general population registries and were included in the current review study. 

From the articles, absolute and relative frequency measures of cancer occurrence were abstracted. The abstracted results were cancer frequency estimates in MPN subtypes, or other subgroups defined by sex, age, length of follow up, etc. Additionally, risks for some specific non-myeloid cancer types were abstracted when available. Either these were high incidence cancers (e.g., breast cancer, colorectal cancer, prostate cancer, and lung cancer) or cancer types that have been reported with an elevated risk in patients with MPN (e.g., skin cancer, kidney cancer, and thyroid cancer). Among the included 12 studies, eight reported on risks of both solid and lymphoid malignancies [9,10,11,13,22,23,28,29], two studies reported on risks of lymphoid malignancies only [24,26], and one study reported on risks of solid cancer only [25] (Table 1 and Table 2). Additionally, one study reporting on prognosis among patients with a second cancer and MPN as the primary cancer compared to matched patients with the same cancers but without MPN was included [27] (Table 3). The studies were published in the period 2009–2018 (Table 1 and Table 2).

From the included studies, effect measures, typically risk estimates, were abstracted. The studies mainly reported the standardized incidence ratio (SIR), the incidence rate ratio (IRR), or the hazard ratio (HR) between patients with MPN and the general population or another comparison group. The SIR is the ratio between the observed number of cancers to the number of cancers expected by age/sex/period-specific incidence rates in the general population [50]. The IRR and the HR are the ratios between either cancer incidences or hazards in the two groups and these two effect measures have a similar interpretation [51].

## 5. Conclusions

In conclusion, our review study reveals that patients with MPN suffer a general increased risk of solid cancers—mainly skin cancer, lung cancer, thyroid cancer, and kidney cancer as well as lymphoid malignancies such as non-Hodgkin lymphoma, chronic lymphocytic leukemia and multiple myeloma. Patients should be advised against smoking because of the increased risk of cancers that are tobacco related, among other good reasons. Although our review confirms that clinicians caring for patients with MPNs should be aware of second cancer risk, the relatively small absolute excess cancer occurrence does not warrant routine clinical surveillance programs in asymptomatic patients but rather watchful clinical follow up. There is a shortage of data that allow for assessment of MPN treatment effects and other factors potentially associated with second cancer risk. Such data could potentially provide more focused and individualized follow-up programs and would be of general interest. For patients with MPN who develop a second cancer, a particular focus on supportive care such as thrombosis prophylaxis remains important.

## Figures and Tables

**Figure 1 cancers-12-03061-f001:**
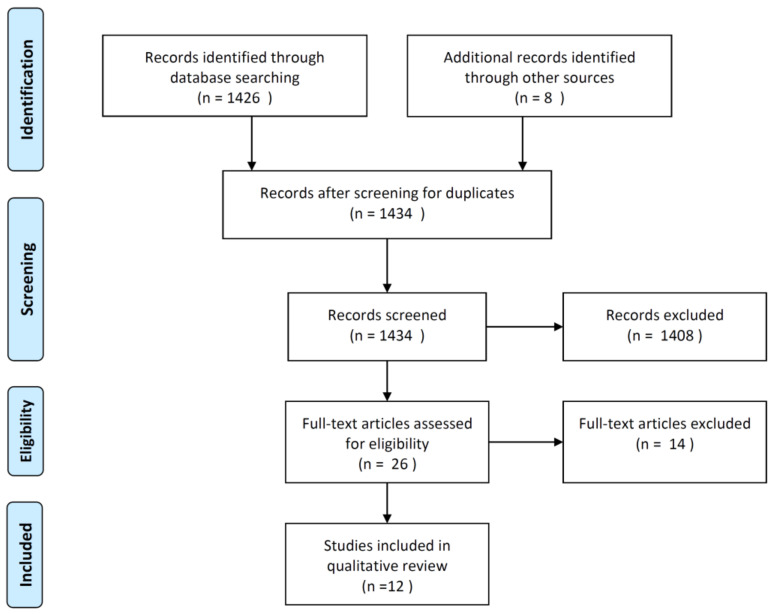
Flow of literature identification, screening, eligibility assessment, and inclusion in our review study of risk of second cancer in patients with chronic myeloproliferative neoplasms. Included studies provided data on cancer occurrence after MPN diagnosis compared either to matched general population controls or to cancer incidence statistics from general population registries.

**Table 1 cancers-12-03061-t001:** Studies on the incidence of solid cancers in patients with chronic myeloproliferative neoplasms (MPNs) compared to cancer incidence in the general population or matched comparisons.

Author	Year	Type of Study	Patient Number	MPN Subtypes	Median Age (Range)	Median Years of Follow Up (Range)	Comparisons	Relative Solid Cancer Occurrence in Patients with MPN vs. Comparisons
Frederiksen [10]	2011	Retrospective cohort	7229	ET (22%) PV (63%) CML (14%)	ET: 65 PV: 66 CML: 60	ET: 4.0 PV: 5.0 CML: 2.4	Incidence of solid neoplasms in the Danish cancer registry	SIR_ET all solid:_ 1.2 (95% CI: 1.0–1.4)
								SIR_PV all solid:_ 3.8 (95% CI: 3.1–4.6)
								SIR_CML all solid:_ 5.2 (95% CI: 2.8–8.7)
								SIR_PV colon_: 0.9 (95% CI: 0.7–1.3)
								SIR_PV lung_: 1.9 (95% CI: 1.6–2.2)
								SIR_PV breast_: 1.1 (95% CI: 0.8–1.5)
								SIR_PV melanoma_: 1.7 (95% CI: 1.0–2.7)
								SIR_PV NMSC_: 1.7 (95% CI: 1.4–1.9)
								SIR_kidney_: 1.9 (95% CI: 1.1–3.0)
								SIR_PV prostate_: 1.3 (95% CI: 1.0–1.6)
Fallah [22]	2011	Retrospective cohort	3530	PV			Incidence of solid neoplasms in the Swedish cancer registry	SIR_colon_: 0.9 (95% CI: 0.6–1.2)
								SIR_lung_: 1.3 (95% CI: 1.0–1.8)
								SIR_breast_: 1.1 (95% CI: 0.8–1.5)
								SIR_melanoma_: 1.9 (95% CI: 1.1–2.9)
								SIR_NMSC_: 2.0 (95% CI: 1.5–2.7)
								SIR_prostate_: 1.0 (95% CI: 0.8–1.2)
								SIR_kidney_: 2.2 (95% CI: 1.5–3.3)
								SIR_thyroid_: 0.5 (95% CI: 0.0–2.7)
Susini [25]	2012	Retrospective cohort	733	ET (51%) PV (41%) PMF (8%)		Mean 6.5	General population incidence of solid neoplasms in the Tuscany cancer registry, Italy	**All solid cancers**
								SIR_MPN:_ 0.9 (95% CI: 0.6–1.1)
								SIR_ET:_ 0.8 (95% CI: 0.5–1.3)
								SIR_PV:_ 1.0 (95% CI: 0.6–1.4)
								**All MPNs**
								SIR_colon_: 0.6 (95% CI: 0.3–1.5)
								SIR_lung_: 0.9 (95% CI: 0.4–1.9)
								SIR_breast_: 0.8 (95% CI: 0.3–1.9)
								SIR_melanoma_: 3.7 (95% CI: 1.4–9.6)
								SIR_NMSC_: 1.0 (95% CI: 0.6–1.9)
								SIR_kidney_: 1.1 (95% CI: 0.3–4.5)
								SIR_prostate_: 0.7 (95% CI: 0.3–1.8)
Khanal [23]	2015	Retrospective cohort	3941	PV	64	4.7 (0.4–12.5)	Incidence of solid neoplasms in the US National Cancer Institute database	SIR_all solid_: 1.2 (95% CI: 1.3–20.7)
								SIR_colon_: 0.6 (95% CI: 0.3–1.0)
								SIR_lung_: 1.2 (95% CI: 0.9–1.6)
								SIR_breast_: 1.0 (95% CI: 0.6–1.5)
								SIR_melanoma_: 1.8 (95% CI: 1.1–2.8)
								SIR_prostate_: 1.3 (95% CI: 1.0–1.7)
								SIR_kidney_: 1.8 (95% CI: 1.0–3.1)
								SIR_thyroid_: 3.1 (95% CI: 1.5–5.7)
Brunner [9]	2016	Retrospective cohort	20,250	ET PV MF MPN-U	ET: 66 PV: 63 MF: 69 MPN-U: 71	3.5	Age-adjusted general US population	**All solid cancers**
								IRR_ET_: 1.4 (95% CI: 1.3–1.5)
								IRR_PV_: 1.4 (95% CI: 1.3–1.6)
								IRR_MF_: 1.5 (95% CI: 1.3–1.8)
								IRR_MPN-U_: 1.6 (95% CI: 1.4–1.8)
								**All MPNs**
								IRR_colon_: 1.0 (95% CI: 0.8–1.1)
								IRR_lung_: 1.6 (95% CI: 1.4–1.8)
								IRR_breast_: 1.1 (95% CI: 0.9–1.3)
								IRR_melanoma_: 2.2 (95% CI: 1.8–2.7)
								IRR_prostate_: 0.6 (95% CI: 0.5–0.6)
								IRR_kidney_: 2.0 (95% CI: 1.6–2.6)
								IRR_thyroid_: 2.3 (95% CI: 1.6–3.2)
Shrestha [28]	2016	Retrospective cohort	8116	ET	68 (1–107)	3 (0.5–10.8)	US SEER database cancer incidence	SIR_all solid_: 1.2 (95% CI: 1.0–1.4)
								SIR_colon_: 0.7 (95% CI: 0.3–1.2)
								SIR_lung_: 1.3 (95% CI: 0.9–1.9)
								SIR_breast_: 1.1 (95% CI: 0.7–1.6)
								SIR_prostate_: 1.1 (95% CI: 0.7–1.7)
								SIR_kidney_: 2.4 (95% CI: 1.2–4.4)
Masarova [29]	2016	Retrospective cohort	417	ET (40%) PV (60%)	(15–84)		US general population	SIR_colon_: 1.6 (95% CI: −0.2–3.5)
								SIR_lung_: 1.2 (95% CI: −2.1–4.4)
								SIR_breast_: 1.3 (95% CI: 0.0–2.5)
								SIR_melanoma_: 3.3 (95% CI: −0.4–7.1)
								SIR_prostate_: 0.8 (95% CI:−0.3–1.9)
								SIR_thyroid_: 3.7 (95% CI: −1.4–8.8)
Landtblom [11]	2018	Retrospective single cohort and matched cohort study	9379	ET (28%) PV (45%) PMF (15%) MPN-NOS (12%)	67.5	7.7	Incidence of solid neoplasms in the Swedish cancer registry For the matched cohort study, four comparisons per MPN patient from the general population matched on age, sex, and residency	**All solid cancers**
								SIR_MPN_: 1.4 (95% CI: 1.3–1.5)
								HR_MPN_: 1.6 (95% CI: 1.5–1.7)
								HR_ET_: 1.6 (95% CI: 1.4–1.8)
								HR_PV_: 1.5 (95% CI: 1.4–1.7)
								HR_PMF_: 1.5 (95% CI: 1.2–1.9)
								HR_MPN-U_: 1.9 (95% CI: 1.5–2.5)
								**All MPNs**
								SIR_colon_: 1.0 (95% CI: 0.8–1.2)
								SIR_lung_: 1.3 (95% CI: 1.1–1.6)
								SIR_breast_: 1.0 (95% CI: 0.8–1.2)
								SIR_melanoma_: 1.9 (95% CI: 1.4–2.7)
								SIR_NMSC_: 3.3 (95% CI: 2.9–3.8)
								SIR_prostate_: 1.1 (95% CI: 1.0–1.3)
								SIR_kidney_: 2.0 (95% CI: 1.5–2.7)
Chattopadhyay [13]	2018	Retrospective cohort	13,805	ET (30%) PV (48%) MF (11%) MPN-NOS (12%)		ET: 4 PV: 6 MF: 2 MPN-U: 3	Incidence of solid neoplasms in the Swedish cancer registry	**All MPNs**
								IRR_colon_: 0.9 (95% CI: 0.8–1.0)
								IRR_lung_: 1.4 (95% CI: 1.2–1.6)
								IRR_breast_: 1.0 (95% CI: 0.8–1.2)
								IRR_melanoma_: 1.8 (95% CI: 1.4–2.2)
								IRR_NMSC_: 2.0 (95% CI: 1.7–2.3)
								IRR_prostate_: 1.1 (95% CI: 1.0–1.2)
								IRR_kidney_: 2.1 (95% CI: 1.7–2.7)

ET: essential thrombocythemia, PV: polycythemia vera, CML: chronic myeloid leukemia, PMF: primary myelofibrosis, SMF: secondary myelofibrosis, MF: myelofibrosis, MPN-U: MPN unspecified, SIR: standardized incidence ratio, NHL: non-Hodgkin lymphoma, NMSC: non-melanoma skin cancer, IRR: incidence rate ratio, and 95% CI: 95% confidence interval. HR: hazard ratio for a solid cancer diagnosis in patients with MPN compared to age, sex, and residency-matched comparisons from the general population. SEER: surveillance, epidemiology, and end results.

**Table 2 cancers-12-03061-t002:** Studies on incidence of malignant lymphoid neoplasms in patients with chronic myeloproliferative neoplasms (MPNs) compared to cancer incidence in the general population or matched comparisons.

Author	Year	Type of Study	Patient Number	MPN Subtypes	Median Age (Range)	Median Years of Follow Up (Range)	Comparisons	Relative Lymphoid Cancer Occurrence in Patients with MPN vs. Comparisons
Vannuchi [26]	2009	Retrospective cohort	820	ET (57%) PV (43%)		3.3	General population incidence of lymphoid neoplasms in the Tuscany cancer registry, Italy	SIR_all_: 3.4 (95% CI: 1.9–6.2)
								SIR_ET_: 2.5 (95% CI: 0.9–6.5)
								SIR_PV_: 4.5 (95% CI: 2.1–9.3)
								SIR_all NHL_: 3.4 (95% CI: 1.4–8.3)
								SIR_all CLL_: 12.4 (95% CI: 4.7–33.1)
Rumi [24]	2011	Retrospective cohort	1915	ET (44%) PV (34%) PMF (18%) SMF (4%)	55.6	5.2 (0–33)	General population incidence of lymphoid neoplasms in the north Italian cancer registry	SIRall: 2.8 (95% CI: 1.8–4.3)
								SIR♂: 3.2 (95% CI: 1.9–5.6)
								SIR♀: 2.2 (95% CI: 1.1–4.7)
								SIR < 50: 6.2 (95% CI: 3.2–11.8)
								SIR + 50: 1.9 (95% CI: 1.1–3.5)
Frederiksen [10]	2011	Retrospective cohort	4625	PV	66	5.0	General population incidence of lymphoid neoplasms in the Danish cancer registry	SIR_NHL_: 1.8 (95% CI: 1.1–2.7)
Fallah [22]	2011	Retrospective cohort	3530	PV			General population incidence of lymphoid neoplasms in the Swedish cancer registry	SIR_NHL_: 1.2 (95% CI: 0.7–2.0)
								SIR_MM_: 1.6 (95% CI: 0.8–3.0)
Khanal [23]	2015	Retrospective cohort	3941	PV	64	4.7 (0.4–12.5)	General population incidence of lymphoid neoplasms in the US National Cancer Institute database	SIR_NHL_: 1.1 (95% CI: 0.6–1.9)
Brunner [9]	2016	Retrospective cohort	20,250	ET, PV, MF, MPN-NOS	ET: 66 PV: 63 MF: 69 MPN-NOS: 71	3.5	General US population lymphoid neoplasm incidences	IRR_NHL_: 2.3 (95% CI: 1.9–2.8)
								IRR_HL_: 3.1 (95% CI: 1.4–6.7)
								IRR_CLL_: 2.5 (95% CI: 1.6–3.6)
								IRR_MM_: 1.6 (95% CI: 1.0–2.3)
Shrestha [28]	2016	Retrospective cohort	8116	ET	68 (1–107)	3 (0.5–10.8)	US SEER database cancer incidence	SIR_lymphoma_: 1.6 (95% CI: 0.8–2.8)
Masarova [29]	2016	Retrospective cohort	417	ET (40%) PV (60%)	(15–84)		US general population	SIR_NHL_: 9.7 (95% CI: 3.0–16.0)
Landtblom [11]	2018	Retrospective single cohort and matched cohort study	9379	ET (28%) PV (45%) PMF (15%) MPN-NOS (12%)	67.5	7.7	General population incidence of lymphoid neoplasms in the Swedish cancer registry. For the matched cohort study, four comparisons per MPN patient from the general population matched on age, sex, and residency	**Lymphoma**
								SIR_MPN_: 2.1 (95% CI: 1.7–2.6)
								HR_MPN_: 2.6 (95% CI: 2.0–3.3)
								HR_ET_: 2.3 (95% CI: 1.3–3.9)
								HR_PV_: 1.9 (95% CI: 1.3–2.8)
								HR_PMF_: 6.0 (95% CI: 3.4–10.8)
								**Multiple Myeloma**
								SIR_MPN_ 1.2 (95% CI: 0.7–1.9)
								HR_MPN_: 1.7 (95% CI: 1.0–3.0)
								HR_ET_: 1.4 (95% CI: 0.5–3.6)
								HR_PV_: 1.6 (95% CI: 0.7–3.5)
								HR_PMF_: 9.0 (95% CI: 1.8–44.0)
Chattopadhyay [13]	2018	Retrospective cohort	13,805	ET (30%) PV (48%) MF (11%) MPN-NOS (12%)		ET: 4 PV: 6 MF: 2 MPN-U: 3	Incidence of lymphoid neoplasms in the Swedish cancer registry	IRR_NHL_: 1.6 (95% CI: 1.3–2.0)
								IRR_HL_: 2.8 (95% CI: 1.5–5.2)
								IRR_MM_: 1.6 (95% CI: 1.1–2.3)

ET: essential thrombocythemia, PV: polycythemia vera, PMF: primary myelofibrosis, SMF: secondary myelofibrosis, MF: myelofibrosis, MPN-NOS: MPN-not otherwise specified, SIR: standardized incidence ratio, NHL: non-Hodgkin lymphoma, HL: Hodgkin lymphoma, MM: multiple myeloma, IRR: incidence rate ratio, and CLL: chronic lymphocytic leukemia. HR: hazard ratio for a lymphoid cancer diagnosis in patients with MPN compared to age, sex, and residency-matched comparisons from the general population. SEER: surveillance, epidemiology, and end results.

**Table 3 cancers-12-03061-t003:** Prognosis in patients with chronic myeloproliferative neoplasms (MPNs) after diagnosis of a new primary solid cancer.

Author	Year	Type of Study	Patient Number	MPN Subtypes	Median Age at Solid Cancer Diagnosis	Median Lag Time to Solid Cancer after MPN Diagnosis (IQR)	Comparisons	Main Findings
Frederiksen [27]	2015	Retrospective cohort	1246	ET (24%) PV (67%) CML (9%)	ET: 73.6 PV: 72.1 CML: 63.7	2.0 (0.8-4-0)	Age/sex-matched patients with the same solid cancers but without preceding MPN (*n* = 5155)	**Five-year cumulative incidence of solid cancer**
								ET: 7.8% (95% CI: 6.6–9.1)
								PV: 6.8% (95% CI: 6.1–7.6)
								CML: 4.7% (95% CI: 3.5–6.1)
								**Death < 5 years**
								HR_ET_: 1.5 (95% CI: 1.2–2.0)
								HR_PV_: 1.2 (95% CI: 1.0–1.4)
								HR_CML_: 1.5 (95% CI: 1.0–2.5)
								**Death ≥ 5 years**
								HR_ET_: 1.7 (95% CI: 1.2–2.4)
								HR_PV_: 1.5 (95% CI: 1.3–1.7)
								HR_CML_: 2.3 (95% CI: 1.3–3.9)

ET: essential thrombocythemia, PV: polycythemia vera, CML: chronic myeloid leukemia, 95% CI: 95% confidence interval, and HR: hazard ratio for death in patients with a solid cancer preceded by a MPN compared to patients with the same solid cancer but without concurrent MPN.

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
