# Peer review of "Risks of Solid and Lymphoid Malignancies in Patients with Myeloproliferative Neoplasms: Clinical Implications"

_cancers, 2020, doi:10.3390/cancers12103061_

Round 1

Reviewer 1 Report

The manuscript by Brabrand and Frederiksen reviews the literature regarding secondary malignancies in myeloproliferative dieases (MPN). Most of the 12 cited studies report on an increase in cancer incidence in MPN. However, the value of this finding remains questionable as there is only little information about the treatment chosen for the treatment of the respective MPN. The potential of drugs (e.g. ruxolitinib) to increase frequency of lymphoid neoplasms is being controversely discussed at the moment for instance, and therefore incidence numbers without referral to the chosen treatment in those patients is less relevant. Increase of solid cancers in this cohort has also been discribed without a correlation to therapy.

I would reject the manuscript as the the information retrieved by the review is fragmentary especially due to the lack of treatment information. The reported conclusions would not change antything in clinical routine from my point of view.

Author Response

The manuscript by Brabrand and Frederiksen reviews the literature regarding secondary malignancies in myeloproliferative dieases (MPN). Most of the 12 cited studies report on an increase in cancer incidence in MPN. However, the value of this finding remains questionable as there is only little information about the treatment chosen for the treatment of the respective MPN. The potential of drugs (e.g. ruxolitinib) to increase frequency of lymphoid neoplasms is being controversely discussed at the moment for instance, and therefore incidence numbers without referral to the chosen treatment in those patients is less relevant. Increase of solid cancers in this cohort has also been discribed without a correlation to therapy.
I would reject the manuscript as the the information retrieved by the review is fragmentary especially due to the lack of treatment information. The reported conclusions would not change antything in clinical routine from my point of view.
Response:
Please see our response to reviewer 2.

Response: We thank the reviewer for the overall positive evaluation. We agree that risk factors potentially associated with second cancer risk is important to identify. We devote a large part of the discussion to discuss previous findings of potential risk factors from other studies. None of these studies however, met our inclusion criteria since they did not provide a comparison group without MPN. The scientific community and clinicians are probably particularly interested in effects of MPN treatment. While it is relatively clear that treatment that are no longer in general use such 32P and pipobroman are carcinogenic, the possible deleterious effects of more recently applied MPN treatments are not straight forward. Among the studies that met inclusion criteria for the review, only two report on data of treatment and find either no or inconclusive effects on cancer risk. Additionally, three studies have reported on cancer risk according to period of diagnosis. If found periodic differences could indirectly suggest effects of changes in management
in the past decades. In order to summarize these other potential risk factors reported in the included studies we have added a new subsection line 177-183. The subsection reads:

“2.7. Other factors potentially associated with second cancer risk

In most of the included studies, data were based on routine registrations and detailed clinical or paraclinical information such as genetic aberrations and treatment were usually not available. However, two studies included different MPN treatments 25,29 or JAK2V617F mutational status 25 in their analyses but did not find them to be associated with second cancer risk. Similarly, period of MPN diagnosis was also not associated with second cancer risk in three other studies 10,11,13.”

We have already report on effects of age, which is an important second cancer risk factor. In order to emphasize this we have changed the heading of subsection 2.6 (line 161) from “Effects of age” to “Effects of age on second cancer risk”.

We also agree that studies that allow for addressing effects of e.g. treatment remain important although such data are generally lacking. In order to emphasize this we have therefore added the following to the conclusion (line 337-339):

“There is a shortage of data that allow for assessment of MPN treatment effects and other factors potentially associated with second cancer risk. Such data could potentially provide more focused and individualized follow-up programmes and would be of general interest. “

Reviewer 2 Report

The study was conducted by experts and is a review of the literature on the relationships between classical myeloproliferative diseases and solid tumors. The selection of studies involved articles comparing cancer incidences with those of the general population. It has been confirmed that MPNs are associated with an excess risk of lymphomas and solid cancers and have reported the risk factors that have been identified in the various studies. The biological plausibility of this association was discussed and, in the end, proposals were made for rules of conduct for the surveillance of these patients.

Comment

The work is interesting and conducted and discussed with great rigor and, in a review like this one, we would have expected that a part of the study would be devoted to the search for risk factors in the individual studies examined. This could have recognized subgroups of MPNs to compare with the general population. Probably this was not possible for all the selected articles, but I would suggest that the AA recommend to the scientific community that,  in the work on MPN and Cancer, the potential risk factors should be detailed which would then allow a meta-analysis of the results.

Author Response

The study was conducted by experts and is a review of the literature on the relationships between classical myeloproliferative diseases and solid tumors. The selection of studies involved articles comparing cancer incidences with those of the general population. It has been confirmed that MPNs are associated with an excess risk of lymphomas and solid cancers and have reported the risk factors that have been identified in the various studies. The biological plausibility of this association was discussed and, in the end, proposals were made for rules of conduct for the surveillance of these patients.
Comment
The work is interesting and conducted and discussed with great rigor and, in a review like this one, we would have expected that a part of the study would be devoted to the search for risk factors in the individual studies examined. This could have recognized subgroups of MPNs to compare with the general population. Probably this was not possible for all the selected articles, but I would suggest that the AA recommend to the scientific community that, in the work on MPN and Cancer, the potential risk factors should be detailed which would then allow a meta-analysis of the results.
Response: We thank the reviewer for the overall positive evaluation. We agree that risk factors potentially associated with second cancer risk is important to identify. We devote a large part of the discussion to discuss previous findings of potential risk factors from other studies. None of these studies however, met our inclusion criteria since they did not provide a comparison group without MPN. The scientific community and clinicians are probably particularly interested in effects of MPN treatment. While it is relatively clear that treatment that are no longer in general use such 32P and pipobroman are carcinogenic, the possible deleterious effects of more recently applied MPN treatments are not straight forward. Among the studies that met inclusion criteria for the review, only two report on data of treatment and find either no or inconclusive effects on cancer risk. Additionally, three studies have reported on cancer risk according to period of diagnosis. If found periodic differences could indirectly suggest effects of changes in management
in the past decades. In order to summarize these other potential risk factors reported in the included studies we have added a new subsection line 177-183. The subsection reads:

“2.7. Other factors potentially associated with second cancer risk

In most of the included studies, data were based on routine registrations and detailed clinical or paraclinical information such as genetic aberrations and treatment were usually not available. However, two studies included different MPN treatments 25,29 or JAK2V617F mutational status 25 in their analyses but did not find them to be associated with second cancer risk. Similarly, period of MPN diagnosis was also not associated with second cancer risk in three other studies 10,11,13.”

We have already report on effects of age, which is an important second cancer risk factor. In order to emphasize this we have changed the heading of subsection 2.6 (line 161) from “Effects of age” to “Effects of age on second cancer risk”.

We also agree that studies that allow for addressing effects of e.g. treatment remain important although such data are generally lacking. In order to emphasize this we have therefore added the following to the conclusion (line 337-339):

“There is a shortage of data that allow for assessment of MPN treatment effects and other factors potentially associated with second cancer risk. Such data could potentially provide more focused and individualized follow-up programmes and would be of general interest. “

Reviewer 3 Report

This is a nice overview on the risk of patients with MPN to develop solid cancers and lymphomas. The high number of MPN patients included in this study is a solid basis to draw conclusions which are clinically relevant for physicians treating such patients. Overall the paper is clearly written and gives concise information on this topic.

Minor comments

A paper from Porpaczy et al reporting an increased incidence of lymhoid malignancies in MF patients treated with ruxolitinib should be mentioned (Aggressive B-cell lymphomas in patients with myelofibrosis receiving JAK1/2 inhibitor therapy. Porpaczy E, et al. Blood. 2018. PMID: 29907599)

The statstical methods used in this study should be described in more detail (for instance SIR and IRR should be explained in the Methods section)

The sentence in line 250 is for me kind of confusing. Smokers and patients with lung disease may develop blood test results (that?) may resemble test results from 251 untreated patients with MPNs such as high haematocrit, thrombocytosis and leucocytosis

Author Response

This is a nice overview on the risk of patients with MPN to develop solid cancers and lymphomas. The high number of MPN patients included in this study is a solid basis to draw conclusions which are clinically relevant for physicians treating such patients. Overall the paper is clearly written and gives concise information on this topic.

Response: We thank the reviewer for the overall positive evaluation.
Minor comments
A paper from Porpaczy et al reporting an increased incidence of lymhoid malignancies in MF patients treated with ruxolitinib should be mentioned (Aggressive B-cell lymphomas in patients with myelofibrosis receiving JAK1/2 inhibitor therapy. Porpaczy E, et al. Blood. 2018. PMID: 29907599)
Response:
We agree that this study should be mentioned in our study. In the discussion line 243-247 we have added the following sentences:
“Recently a study including two cohorts of 626 and 929 patients with MPN observed that patients treated with JAK1/2 inhibitors developed aggressive lymphomas (n=6) 15-16 times more frequently than patients not exposed to these treatments 35. Due to the small numbers of patients who developed lymphoma, risk estimates were imprecise 35.”

The statstical methods used in this study should be described in more detail (for instance SIR and IRR should be explained in the Methods section)
Response:
We agree that summarizing the abstracted type of effect estimates from the included studies should be provided in the methods sections. We have therefore added the following paragraph to the methods section (line 339-345).
“From the included studies, effect measures typically risk estimates was abstracted. The studies mainly reported the standardized incidence ratio (SIR), the incidence rate ratio (IRR), or the hazard ratio (HR) between patients with MPN and the general population or another comparison group. The SIR is the ratio between the observed number of cancers to the number of cancers expected by age-sex-period specific incidence rates in the general population 50. The IRR and the HR are the ratios between either cancer incidences or hazards in the two groups and these two effect measures have a similar interpretation 51.

The sentence in line 250 is for me kind of confusing. Smokers and patients with lung disease may develop blood test results (that?) may resemble test results from 251 untreated patients with MPNs such as high haematocrit, thrombocytosis and leukocytosis
Response:
Thank you, for pointing this out. We have amended the missing “that”. The sentence now reads (line 261263):
“Smokers and patients with lung disease may develop blood test results that may resemble test results from untreated patients with MPNs …”

Reviewer 4 Report

In this article authors would like to study the risk to developed solid and lymphoid tumors in patients with myeloproliferative neoplasm (MN).

Major Comments

The results obtained in the review are relevant but not especially novel.

To improve the article, I considered relevant to include data that could selected which patients present higher risk to developed solid or lymphoid tumors.

Treatment received, familiar history, genetic risk factors (variants correlated with genomic instability). I suppose that is complicated to obtain the information of all cases (65.000) but I think is important try to get in a proportion of them.

Furthermore, I also think that in terms of surveillance you must be more specific. How long clinicians must control patients? How often? What test should be performed? 

Minor Comments

  • Page 2, Line 42. Include CML.
  • Figure 1. In the flow literature identification, you must explain with more detail the reasons why patients have been excluded.
  • Line 66. In sentence “Some patients are likely to have been reported in multiple publications…Can you estimate how many?
  • Add and discus new publications (Maria Morchetti et al, 2019; De Stefano V Blood 2019).

Author Response

In this article authors would like to study the risk to developed solid and lymphoid tumors in patients with myeloproliferative neoplasm (MN).
Major Comments
The results obtained in the review are relevant but not especially novel.
To improve the article, I considered relevant to include data that could selected which patients present higher risk to developed solid or lymphoid tumors.
Response:
We agree that risk factors associated with a particularly increased risk of second cancer is of general interest. We have emphasized the risk factors that are reported in our included studies in the subsections 2.6 and 2.7. Please see also our response to reviewer 2.
Treatment received, familiar history, genetic risk factors (variants correlated with genomic instability). I suppose that is complicated to obtain the information of all cases (65.000) but I think is important try to get in a proportion of them.
Response:
We agree that it would be a great benefit if detailed patient characteristics such as treatment data, results from genetic analyses etc. were reported to the registries that formed the basis for almost all the included studies. As we mention in our reply to reviewer 2 only two studies study included analyses of treatment or genetic data. These additional results have been added in subsection 2.7. In general large clinical databases that also include detailed treatment information remains an important resource to study short-term and long-term outcomes in MPNs. However, it is both laborious and time consuming to construct and report to such a database. In Denmark, reporting to the “Danish Chronic Myeloid Neoplasia Registry” (DCMR) that include such details started in 2010. Analysis of data for research purposes in the DCMR has just recently started. Until databases such as the DCMR are “mature” for analyses of long-term outcomes - e.g. second cancer - routine registry databases with long and often complete follow-up like the ones included in the reviewed studies are very valuable resources for research although they cannot answer all questions. It is beyond the scope and possibilities of this review study to retrieve original patient data from multiple registries in different countries.
Furthermore, I also think that in terms of surveillance you must be more specific. How long clinicians must control patients? How often? What test should be performed?
Response:
We agree that specific guidelines about which patients that should be investigated more and when are of interest but results from the included studies does not allow for firm conclusions and exact recommendations. However, in our conclusion we do suggest that routine surveillance in asymptomatic patients is unlikely to provide a survival benefit based on the small absolute differences in cancer incidence among patients with MPNs and the general population. Additionally, in our subsection 3.1 we summarize the subgroups and second cancers that are particularly relevant to keep in clinical focus. We believe that until more solid data are available this information combined with the absolute risk measures that we also provide are helpful for clinicians or institutions to decide on standards for follow-up among patients with MPNs.
Minor Comments
ï‚· Page 2, Line 42. Include CML. Response: We have included CML as suggested. Line 42-44 now reads:

“Myeloproliferative neoplasms (MPNs) encompassing essential thrombocythaemia (ET), polycythaemia vera (PV), primary myelofibrosis (PMF), chronic myeloid leukemia (CML), and unspecified MPNs are chronic stem cell cancers of the bone marrow 1.”

ï‚· Figure 1. In the flow literature identification, you must explain with more detail the reasons why patients have been excluded.

Response: In Figure 1 we summarize the results of literature search and the review process. We did not exclude specific patients. We agree that details of the selection process should be included in the figure legend. We have therefore added the criteria for study inclusion in the figure legend which now reads (line 324-327).
“Figure 1. Flow of literature identification, screening, eligibility assessment, and inclusion in review study of risk of second cancer in patients with chronic myeloproliferative neoplasms. Included studies provided data on cancer occurrence after MPN diagnosis compared either to matched general population controls or to cancer incidence statistics from general population registries.”

ï‚· Line 66. In sentence “Some patients are likely to have been reported in multiple publications…Can you estimate how many? Response:
Unfortunately, it is not possible to give an accurate estimate of the overlap in populations included in the three different studies. However, in order to provide some detail of the overlapping study populations the three study periods have been added. The sentence now reads (line 64-66):

“The 12 studies identified included analysis of more than 65,000 patients with MPNs and various population comparison groups. Three studies identified and followed patients through the Swedish cancer registry that were diagnosed with MPN 1958-2006 22, 1973-2009 11, or 1958-2015 13 and therefore some patients are likely to have been reported in multiple publications and analyses. “

ï‚· Add and discus new publications (Maria Morchetti et al, 2019; De Stefano V Blood 2019). Response:
For the first study, we believe the reviewer refers to the study by Monia Marchetti et al published in. Am J Hematol. 2020 Mar;95(3):295-301. This is a follow-up study on the study mentioned below. This new study is devoted to study prognosis in MPN + second cancer. We have added this study in our discussion of prognosis of MPN+second cancer (line 290-292).

“Intriguingly, a recent study suggest that active MPN treatment may affect survival negatively in second cancer, however, the mechanism e.g. suboptimal second cancer management is unknown 48.”

For the second study we did not find a publication of De Stefano-V in Blood. We did find a publication in Leukemia from 2019 with De Stefano-V as a co-author. We already discuss this study and publication in our review (reference 20). Please direct us more specifically to the study mentioned by the reviewer in case we overlooked a relevant reference.

Round 2

Reviewer 1 Report

The authors have adressed the reviewers issues adequately, the paper can therefore be accepted.